# Exponential Family Embeddings

**Maja Rudolph**
Columbia University

**Francisco J. R. Ruiz**
Univ. of Cambridge
Columbia University

**Stephan Mandt**
Columbia University

**David M. Blei**
Columbia University

## Abstract

Word embeddings are a powerful approach for capturing semantic similarity among terms in a vocabulary. In this paper, we develop *exponential family embeddings*, a class of methods that extends the idea of word embeddings to other types of high-dimensional data. As examples, we studied neural data with real-valued observations, count data from a market basket analysis, and ratings data from a movie recommendation system. The main idea is to model each observation conditioned on a set of other observations. This set is called the context, and the way the context is defined is a modeling choice that depends on the problem. In language the context is the surrounding words; in neuroscience the context is close-by neurons; in market basket data the context is other items in the shopping cart. Each type of embedding model defines the context, the exponential family of conditional distributions, and how the latent embedding vectors are shared across data. We infer the embeddings with a scalable algorithm based on stochastic gradient descent. On all three applications—neural activity of zebrafish, users' shopping behavior, and movie ratings—we found exponential family embedding models to be more effective than other types of dimension reduction. They better reconstruct held-out data and find interesting qualitative structure.

## 1 Introduction

Word embeddings are a powerful approach for analyzing language (Bengio et al., 2006; Mikolov et al., 2013a,b; Pennington et al., 2014). A word embedding method discovers distributed representations of words; these representations capture the semantic similarity between the words and reflect a variety of other linguistic regularities (Rumelhart et al., 1986; Bengio et al., 2006; Mikolov et al., 2013c). Fitted word embeddings can help us understand the structure of language and are useful for downstream tasks based on text.

There are many variants, adaptations, and extensions of word embeddings (Mikolov et al., 2013a,b; Mnih and Kavukcuoglu, 2013; Levy and Goldberg, 2014; Pennington et al., 2014; Vilnis and Mc-Callum, 2015), but each reflects the same main ideas. Each term in a vocabulary is associated with two latent vectors, an *embedding* and a *context vector*. These two types of vectors govern conditional probabilities that relate each word to its surrounding context. Specifically, the conditional probability of a word combines its embedding and the context vectors of its surrounding words. (Different methods combine them differently.) Given a corpus, we fit the embeddings by maximizing the conditional probabilities of the observed text.

In this paper we develop the *exponential family embedding (EF-EMB)*, a class of models that generalizes the spirit of word embeddings to other types of high-dimensional data. Our motivation is that other types of data can benefit from the same assumptions that underlie word embeddings, namely that a data point is governed by the other data in its context. In language, this is the foundational idea that words with similar meanings will appear in similar contexts (Harris, 1954). We use the tools of exponential families (Brown, 1986) and generalized linear models (GLMs) (McCullagh and Nelder, 1989) to adapt this idea beyond language.

As one example beyond language, we will study computational neuroscience. Neuroscientists measure sequential neural activity across many neurons in the brain. Their goal is to discover patterns in these data with the hope of better understanding the dynamics and connections among neurons. In this example, a context can be defined as the neural activities of other nearby neurons, or as neural activity in the past. Thus, it is plausible that the activity of each neuron depends on its context. We will use this idea to fit latent embeddings of neurons, representations of neurons that uncover hidden features which help suggest their roles in the brain.

Another example we study involves shoppers at the grocery store. Economists collect shopping data (called "market basket data") and are interested in building models of purchase behavior for downstream econometric analysis, e.g., to predict demand and market changes. To build such models, they seek features of items that are predictive of when they are purchased and in what quantity. Similar to language, purchasing an item depends on its context, i.e., the other items in the shopping cart. In market basket data, Poisson embeddings can capture important econometric concepts, such as items that tend not to occur together but occur in the same contexts (substitutes) and items that co-occur, but never one without the other (complements).

We define an EF-EMB, such as one for neuroscience or shopping data, with three ingredients. (1) We define the *context*, which specifies which other data points each observation depends on. (2) We define the *conditional exponential family*. This involves setting the appropriate distribution, such as a Gaussian for real-valued data or a Poisson for count data, and the way to combine embeddings and context vectors to form its natural parameter. (3) We define the *embedding structure*, how embeddings and context vectors are shared across the conditional distributions of each observation. These three ingredients enable a variety of embedding models.

We describe EF-EMB models and develop efficient algorithms for fitting them. We show how existing methods, such as continuous bag of words (CBOW) (Mikolov et al., 2013a) and negative sampling (Mikolov et al., 2013b), can each be viewed as an EF-EMB. We study our methods on three different types of data—neuroscience data, shopping data, and movie ratings data. Mirroring the success of word embeddings, EF-EMB models outperform traditional dimension reduction, such as exponential family principal component analysis (PCA) (Collins et al., 2001) and Poisson factorization (Gopalan et al., 2015), and find interpretable features of the data.

**Related work.** EF-EMB models generalize CBOW (Mikolov et al., 2013a) in the same way that exponential family PCA (Collins et al., 2001) generalizes PCA, GLMS (McCullagh and Nelder, 1989) generalize regression, and deep exponential families (Ranganath et al., 2015) generalize sigmoid belief networks (Neal, 1990). A linear EF-EMB (which we define precisely below) relates to context-window-based embedding methods such as CBOW or the vector log-bilinear language model (VLBL) (Mikolov et al., 2013a; Mnih and Kavukcuoglu, 2013), which model a word given its context. The more general EF-EMB relates to embeddings with a nonlinear component, such as the skip-gram (Mikolov et al., 2013a) or the inverse vector log-bilinear language model (IVLBL) (Mnih and Kavukcuoglu, 2013). (These methods might appear linear but, when viewed as a conditional probabilistic model, the normalizing constant of each word induces a nonlinearity.)

Researchers have developed different approximations of the word embedding objective to scale the procedure. These include noise contrastive estimation (Gutmann and Hyvärinen, 2010; Mnih and Teh, 2012), hierarchical softmax (Mikolov et al., 2013b), and negative sampling (Mikolov et al., 2013a). We explain in Section 2.2 and Supplement A how negative sampling corresponds to biased stochastic gradients of an EF-EMB objective.

## 2 Exponential Family Embeddings

We consider a matrix $x = x_{1:I}$ of $I$ observations, where each $x_i$ is a $D$-vector. As one example, in language $x_i$ is an indicator vector for the word at position $i$ and $D$ is the size of the vocabulary. As another example, in neural data $x_i$ is the neural activity measured at index pair $i = (n, t)$, where $n$ indexes a neuron and $t$ indexes a time point; each measurement is a scalar ($D = 1$).

The goal of an exponential family embedding (EF-EMB) is to derive useful features of the data. There are three ingredients: a context function, a conditional exponential family, and an embedding structure. These ingredients work together to form the objective. First, the EF-EMB models each data point conditional on its context; the context function determines which other data points are at play. Second,

the conditional distribution is an appropriate exponential family, e.g., a Gaussian for real-valued data. Its parameter is a function of the embeddings of both the data point and its context. Finally, the embedding structure determines which embeddings are used when the $i$th point appears, either as data or in the context of another point. The objective is the sum of the log probabilities of each data point given its context. We describe each ingredient, followed by the EF-EMB objective. Examples are in Section 2.1.

**Context.** Each data point $i$ has a *context* $c_i$, which is a set of indices of other data points. The EF-EMB models the conditional distribution of $x_i$ given the data points in its context.

The context is a modeling choice; different applications will require different types of context. In language, the data point is a word and the context is the set of words in a window around it. In neural data, the data point is the activity of a neuron at a time point and the context is the activity of its surrounding neurons at the same time point. (It can also include neurons at future time or in the past.) In shopping data, the data point is a purchase and the context is the other items in the cart.

**Conditional exponential family.** An EF-EMB models each data point $x_i$ conditional on its context $x_{c_i}$. The distribution is an appropriate exponential family,

$$x_i \mid x_{c_i} \sim \text{ExpFam}(\eta_i(x_{c_i}), t(x_i)), \tag{1}$$

where $\eta_i(x_{c_i})$ is the natural parameter and $t(x_i)$ is the sufficient statistic. In language modeling, this family is usually a categorical distribution. Below, we will study Gaussian and Poisson.

We parameterize the conditional with two types of vectors, embeddings and context vectors. The *embedding* of the $i$th data point helps govern its distribution; we denote it $\rho[i] \in \mathbb{R}^{K \times D}$. The *context vector* of the $i$th data point helps govern the distribution of data for which $i$ appears in their context; we denote it $\alpha[i] \in \mathbb{R}^{K \times D}$.

How to define the natural parameter as a function of these vectors is a modeling choice. It captures how the context interacts with an embedding to determine the conditional distribution of a data point. Here we focus on the *linear embedding*, where the natural parameter is a function of a linear combination of the latent vectors,

$$\eta_i(x_{c_i}) = f_i\left(\rho[i]^\top \sum_{j \in c_i} \alpha[j] x_j\right). \tag{2}$$

Following the nomenclature of generalized linear models (GLMS), we call $f_i(\cdot)$ the *link function*. We will see several examples of link functions in Section 2.1.

This is the setting of many existing word embedding models, though not all. Other models, such as the skip-gram, determine the probability through a "reverse" distribution of context words given the data point. These non-linear embeddings are still instances of an EF-EMB.

**Embedding structure.** The goal of an EF-EMB is to find embeddings and context vectors that describe features of the data. The *embedding structure* determines how an EF-EMB shares these vectors across the data. It is through sharing the vectors that we learn an embedding for the object of primary interest, such as a vocabulary term, a neuron, or a supermarket product. In language the same parameters $\rho[i] = \rho$ and $\alpha[i] = \alpha$ are shared across all positions $i$. In neural data, observations share parameters when they describe the same neuron. Recall that the index connects to both a neuron and time point $i = (n, t)$. We share parameters with $\rho[i] = \rho_n$ and $\alpha[i] = \alpha_n$ to find embeddings and context vectors that describe the neurons. Other variants might tie the embedding and context vectors to find a single set of latent variables, $\rho[i] = \alpha[i]$.

**The objective function.** The EF-EMB objective sums the log conditional probabilities of each data point, adding regularizers for the embeddings and context vectors.[1] We use log probability functions as regularizers, e.g., a Gaussian probability leads to $\ell_2$ regularization. We also use regularizers to constrain the embeddings, e.g., to be non-negative. Thus, the objective is

$$\mathcal{L}(\rho, \alpha) = \sum_{i=1}^{I} \left(\eta_i^\top t(x_i) - a(\eta_i)\right) + \log p(\rho) + \log p(\alpha). \tag{3}$$

We maximize this objective with respect to the embeddings and context vectors. In Section 2.2 we explain how to fit it with stochastic gradients.

Equation (3) can be seen as a likelihood function for a bank of GLMs (McCullagh and Nelder, 1989). Each data point is modeled as a response conditional on its "covariates," which combine the context vectors and context, e.g., as in Equation (2); the coefficient for each response is the embedding itself. We use properties of exponential families and results around GLMs to derive efficient algorithms for EF-EMB models.

## 2.1 Examples

We highlight the versatility of EF-EMB models with three example models and their variations. We develop the Gaussian embedding (G-EMB) for analyzing real observations from a neuroscience application; we also introduce a nonnegative version, the nonnegative Gaussian embedding (NG-EMB). We develop two Poisson embedding models, Poisson embedding (P-EMB) and additive Poisson embedding (AP-EMB), for analyzing count data; these have different link functions. We present a categorical embedding model that corresponds to the continuous bag of words (CBOW) word embedding (Mikolov et al., 2013a). Finally, we present a Bernoulli embedding (B-EMB) for binary data. In Section 2.2 we explain how negative sampling (Mikolov et al., 2013b) corresponds to biased stochastic gradients of the B-EMB objective. For convenience, these acronyms are in Table 1.

| | |
|---|---|
| **EF-EMB** | *exponential family embedding* |
| **G-EMB** | *Gaussian embedding* |
| **NG-EMB** | *nonnegative Gaussian embedding* |
| **P-EMB** | *Poisson embedding* |
| **AP-EMB** | *additive Poisson embedding* |
| **B-EMB** | *Bernoulli embedding* |

**Table 1:** Acronyms used for exponential family embeddings.

**Example 1: Neural data and Gaussian observations.** Consider the (calcium) expression of a large population of zebrafish neurons (Ahrens et al., 2013). The data are processed to extract the locations of the $N$ neurons and the neural activity $x_i = x_{(n,t)}$ across location $n$ and time $t$. The goal is to model the similarity between neurons in terms of their behavior, to embed each neuron in a latent space such that neurons with similar behavior are close to each other.

We consider two neurons similar if they behave similarly in the context of the activity pattern of their surrounding neurons. Thus we define the context for data index $i = (n, t)$ to be the indices of the activity of nearby neurons at the same time. We find the K-nearest neighbors (KNN) of each neuron (using a Ball-tree algorithm) according to their spatial distance in the brain. We use this set to construct the context $c_i = c_{(n,t)} = \{(m, t) | m \in KNN(n)\}$. This context varies with each neuron, but is constant over time.

With the context defined, each data point $x_i$ is modeled with a conditional Gaussian. The conditional mean is the inner product from Equation (2), where the context is the simultaneous activity of the nearest neurons and the link function is the identity. The conditionals of two observations share parameters if they correspond to the same neuron. The embedding structure is thus $\rho[i] = \rho_n$ and $\alpha[i] = \alpha_n$ for all $i = (n, t)$. Similar to word embeddings, each neuron has two distinct latent vectors: the neuron embedding $\rho_n \in \mathbb{R}^K$ and the context vector $\alpha_n \in \mathbb{R}^K$.

These ingredients, along with a regularizer, combine to form a neural embedding objective. G-EMB uses $\ell_2$ regularization (i.e., a Gaussian prior); NG-EMB constrains the vectors to be nonnegative ($\ell_2$ regularization on the logarithm. i.e., a log-normal prior).

**Example 2: Shopping data and Poisson observations.** We also study data about people shopping. The data contains the individual purchases of anonymous users in chain grocery and drug stores. There are $N$ different items and $T$ trips to the stores among all households. The data is a sparse $N \times T$ matrix of purchase counts. The entry $x_i = x_{(n,t)}$ indicates the number of units of item $n$ that was purchased on trip $t$. Our goal is to learn a latent representation for each product that captures the similarity between them.

We consider items to be similar if they tend to be purchased in with similar groups of other items. The *context* for observation $x_i$ is thus the other items in the shopping basket on the same trip. For the purchase count at index $i = (n, t)$, the context is $c_i = \{j = (m, t) | m \neq n\}$.

We use conditional Poisson distributions to model the count data. The sufficient statistic of the Poisson is $t(x_i) = x_i$, and its natural parameter is the logarithm of the rate (i.e., the mean). We set the natural parameter as in Equation (2), with the link function defined below. The embedding structure is the same as in G-EMB, producing embeddings for the items.

We explore two choices for the link function. P-EMB uses an identity link function. Since the conditional mean is the exponentiated natural parameter, this implies that the context items contribute multiplicatively to the mean. (We use $\ell_2$-regularization on the embeddings.) Alternatively, we can constrain the parameters to be nonnegative and set the link function $f(\cdot) = \log(\cdot)$. This is AP-EMB, a model with an additive mean parameterization. (We use $\ell_2$-regularization in log-space.) AP-EMB only captures positive correlations between items.

**Example 3: Text modeling and categorical observations.** EF-EMBS are inspired by word embeddings, such as CBOW (Mikolov et al., 2013a). CBOW is a special case of an EF-EMB; it is equivalent to a multivariate EF-EMB with categorical conditionals. In the notation here, each $x_i$ is an indicator vector of the $i$th word. Its dimension is the vocabulary size. The context of the $i$th word are the other words in a window around it (of size $w$), $c_i = \{j \neq i | i - w \leq j \leq i + w\}$.

The distribution of $x_i$ is categorical, conditioned on the surrounding words $\boldsymbol{x}_{c_i}$; this is a softmax regression. It has natural parameter as in Equation (2) with an identity link function. The embedding structure imposes that parameters are shared across all observed words. The embeddings are shared globally ($\rho[i] = \rho, \alpha[i] = \alpha \in \mathbb{R}^{N \times K}$). The word and context embedding of the $n^{th}$ word is the $n^{th}$ row of $\rho$ and $\alpha$ respectively. CBOW does not use any regularizer.

**Example 4: Text modeling and binary observations.** One way to simplify the CBOW objective is with a model of each entry of the indicator vectors. The data are binary and indexed by $i = (n, v)$, where $n$ is the position in the text and $v$ indexes the vocabulary; the variable $x_{n,v}$ is the indicator that word $n$ is equal to term $v$. (This model relaxes the constraint that for any $n$ only one $x_{n,v}$ will be on.) With this notation, the context is $c_i = \{(j, v') | \forall v', j \neq n, n - w \leq j \leq n + w\}$; the embedding structure is $\rho[i] = \rho[(n, v)] = \rho_v$ and $\alpha[i] = \alpha[(n, v)] = \alpha_v$.

We can consider different conditional distributions in this setting. As one example, set the conditional distribution to be a Bernoulli with an identity link; we call this the B-EMB model for text. In Section 2.2 we show that biased stochastic gradients of the B-EMB objective recovers negative sampling (Mikolov et al., 2013b). As another example, set the conditional distribution to Poisson with link $f(\cdot) = \log(\cdot)$. The corresponding embedding model relates closely to Poisson approximations of distributed multinomial regression (Taddy et al., 2015).

## 2.2 Inference and Connection to Negative Sampling

We fit the embeddings $\rho[i]$ and context vectors $\alpha[i]$ by maximizing the objective function in Equation (3). We use stochastic gradient descent (SGD) with Adagrad (Duchi et al., 2011). We can derive the analytic gradient of the objective function using properties of the exponential family (see the Supplement for details). The gradients linearly combine the data in summations we can approximate using subsampled minibatches of data. This reduces the computational cost.

When the data is sparse, we can split the gradient into the summation of two terms: one term corresponding to all data entries $i$ for which $x_i \neq 0$, and one term corresponding to those data entries $x_i = 0$. We compute the first term of the gradient exactly—when the data is sparse there are not many summations to make—and we estimate the second term by subsampling the zero entries. Compared to computing the full gradient, this reduces the complexity when most of the entries $x_i$ are zero. But it retains the strong information about the gradient that comes from the non-zero entries.

This relates to negative sampling, which is used to approximate the skip-gram objective (Mikolov et al., 2013b). Negative sampling re-defines the skip-gram objective to distinguish target (observed) words from randomly drawn words, using logistic regression. The gradient of the stochastic objective is identical to a noisy but biased estimate of the gradient for a B-EMB model. To obtain the equivalence, preserve the terms for the non-zero data and subsample terms for the zero data. While an unbiased

| Model | single neuron held out | | 25% of neurons held out | |
|---|---|---|---|---|
| | $K = 10$ | $K = 100$ | $K = 10$ | $K = 100$ |
| FA | $0.290 \pm 0.003$ | $0.275 \pm 0.003$ | $0.290 \pm 0.003$ | $0.276 \pm 0.003$ |
| G-EMB (c=10) | $0.239 \pm 0.006$ | $0.239 \pm 0.005$ | $0.246 \pm 0.004$ | $0.245 \pm 0.003$ |
| G-EMB (c=50) | $0.227 \pm 0.002$ | $\mathbf{0.222 \pm 0.002}$ | $0.235 \pm 0.003$ | $\mathbf{0.232 \pm 0.003}$ |
| NG-EMB (c=10) | $0.263 \pm 0.004$ | $0.261 \pm 0.004$ | $0.250 \pm 0.004$ | $0.261 \pm 0.004$ |

**Table 2:** Analysis of neural data: mean squared error and standard errors of neural activity (on the test set) for different models. Both EF-EMB models significantly outperform FA; G-EMB is more accurate than NG-EMB.

stochastic gradient would rescale the subsampled terms, negative sampling does not. Thus, negative sampling corresponds to a biased estimate, which down-weights the contribution of the zeros. See the Supplement for the mathematical details.

# 3 Empirical Study

We study exponential family embedding (EF-EMB) models on real-valued and count-valued data, and in different application domains—computational neuroscience, shopping behavior, and movie ratings. We present quantitative comparisons to other dimension reduction methods and illustrate how we can glean qualitative insights from the fitted embeddings.

## 3.1 Real Valued Data: Neural Data Analysis

**Data.** We analyze the neural activity of a larval zebrafish, recorded at single cell resolution for 3000 time frames (Ahrens et al., 2013). Through genetic modification, individual neurons express a calcium indicator when they fire. The resulting calcium imaging data is preprocessed by a nonnegative matrix factorization to identify neurons, their locations, and the fluorescence activity $x_t^* \in \mathbb{R}^N$ of the individual neurons over time (Friedrich et al., 2015). Using this method, our data contains 10,000 neurons (out of a total of 200,000).

We fit all models on the lagged data $x_t = x_t^* - x_{t-1}^*$ to filter out correlations based on calcium decay and preprocessing.[2] The calcium levels can be measured with great spatial resolution but the temporal resolution is poor; the neuronal firing rate is much higher than the sampling rate. Hence we ignore all "temporal structure" in the data and model the simultaneous activity of the neurons. We use the Gaussian embedding (G-EMB) and nonnegative Gaussian embedding (NG-EMB) from Section 2.1 to model the lagged activity of the neurons conditional on the lags of surrounding neurons. We study context sizes $c \in \{10, 50\}$ and latent dimension $K \in \{10, 100\}$.

**Models.** We compare EF-EMB to probabilistic factor analysis (FA), fitting $K$-dimensional factors for each neuron and $K$-dimensional factor loadings for each time frame. In FA, each entry of the data matrix is Gaussian distributed, with mean equal to the inner product of the corresponding factor and factor loading.

**Evaluation.** We train each model on a random sample of 90% of the lagged time frames and hold out 5% each for validation and testing. With the test set, we use two types of evaluation. (1) *Leave one out*: For each neuron $x_i$ in the test set, we use the measurements of the other neurons to form predictions. For FA this means the other neurons are used to recover the factor loadings; for EF-EMB this means the other neurons are used to construct the context. (2) *Leave 25% out*: We randomly split the neurons into 4 folds. Each neuron is predicted using the three sets of neurons that are out of its fold. (This is a more difficult task.) Note in EF-EMB, the missing data might change the size of the context of some neurons. See Table 5 in Supplement C for the choice of hyperparameters.

**Results.** Table 2 reports both types of evaluation. The EF-EMB models significantly outperform FA in terms of mean squared error on the test set. G-EMB obtains the best results with 100 components and a context size of 50. Figure 1 illustrates how to use the learned embeddings to hypothesize connections between nearby neurons.

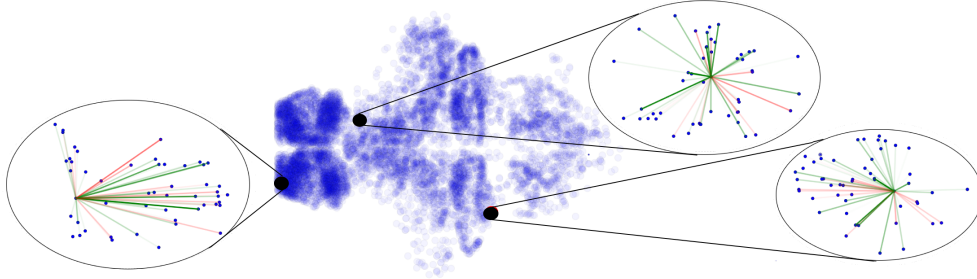

**Figure 1:** Top view of the zebrafish brain, with blue circles at the location of the individual neurons. We zoom on 3 neurons and their 50 nearest neighbors (small blue dots), visualizing the "synaptic weights" learned by a G-EMB model ($K = 100$). The edge color encodes the inner product of the neural embedding vector and the context vectors $\rho_n^\top \alpha_m$ for each neighbor $m$. Positive values are green, negative values are red, and the transparency is proportional to the magnitude. With these weights we can hypothesize how nearby neurons interact.

| Model | $K = 20$ | $K = 100$ | $K = 20$ | $K = 100$ |
|---|---|---|---|---|
| P-EMB | $-7.497 \pm 0.007$ | $-7.199 \pm 0.008$ | $\mathbf{-5.691 \pm 0.006}$ | $-5.726 \pm 0.005$ |
| P-EMB (dw) | $-7.110 \pm 0.007$ | $\mathbf{-6.950 \pm 0.007}$ | $-5.790 \pm 0.003$ | $-5.798 \pm 0.003$ |
| AP-EMB | $-7.868 \pm 0.005$ | $-8.414 \pm 0.003$ | $-5.964 \pm 0.003$ | $-6.118 \pm 0.002$ |
| HPF | $-7.740 \pm 0.008$ | $-7.626 \pm 0.007$ | $-5.787 \pm 0.006$ | $-5.859 \pm 0.006$ |
| Poisson PCA | $-8.314 \pm 0.009$ | $-11.01 \pm 0.01$ | $-5.908 \pm 0.006$ | $-7.50 \pm 0.01$ |

**(a)** Market basket analysis.  **(b)** Movie ratings.

**Table 3:** Comparison of predictive log-likelihood between P-EMB, AP-EMB, hierarchical Poisson factorization (HPF) (Gopalan et al., 2015), and Poisson principal component analysis (PCA) (Collins et al., 2001) on held out data. The P-EMB model outperforms the matrix factorization models in both applications. For the shopping data, downweighting the zeros improves the performance of P-EMB.

### 3.2   Count Data: Market Basket Analysis and Movie Ratings

We study the Poisson models Poisson embedding (P-EMB) and additive Poisson embedding (AP-EMB) on two applications: shopping and movies.

**Market basket data.**   We analyze the IRI dataset[3] (Bronnenberg et al., 2008), which contains the purchases of anonymous households in chain grocery and drug stores. It contains $137, 632$ trips in 2012. We remove items that appear fewer than 10 times, leaving a dataset with $7, 903$ items. The context for each purchase is the other purchases from the same trip.

**MovieLens data.**   We also analyze the MovieLens-$100K$ dataset (Harper and Konstan, 2015), which contains movie ratings on a scale from 1 to 5. We keep only positive ratings, defined to be ratings of 3 or more (we subtract 2 from all ratings and set the negative ones to 0). The context of each rating is the other movies rated by the same user. After removing users who rated fewer than 20 movies and movies that were rated fewer than 50 times, the dataset contains 777 users and 516 movies; the sparsity is about 5%.

**Models.**   We fit the P-EMB and the AP-EMB models using number of components $K \in \{20, 100\}$. For each $K$ we select the Adagrad constant based on best predictive performance on the validation set. (The parameters we used are in Table 5.) In these datasets, the distribution of the context size is heavy tailed. To handle larger context sizes we pick a link function for the EF-EMB model which rescales the sum over the context in Equation (2) by the context size (the number of terms in the sum). We also fit a P-EMB model that artificially downweights the contribution of the zeros in the objective function by a factor of 0.1, as done by Hu et al. (2008) for matrix factorization. We denote it as "P-EMB (dw)."

| Maruchan chicken ramen | Yoplait strawberry yogurt | Mountain Dew soda | Dean Foods 1 % milk |
|---|---|---|---|
| M. creamy chicken ramen | Yoplait apricot mango yogurt | Mtn. Dew orange soda | Dean Foods 2 % milk |
| M. oriental flavor ramen | Yoplait strawberry orange smoothie | Mtn. Dew lemon lime soda | Dean Foods whole milk |
| M. roast chicken ramen | Yoplait strawberry banana yogurt | Pepsi classic soda | Dean Foods chocolate milk |

**Table 4:** Top 3 similar items to a given example query words (bold face). The P-EMB model successfuly captures similarities.

We compare the predictive performance with HPF (Gopalan et al., 2015) and Poisson PCA (Collins et al., 2001). Both HPF and Poisson PCA factorize the data into $K$-dimensional positive vectors of user preferences, and $K$-dimensional positive vectors of item attributes. AP-EMB and HPF parameterize the mean additively; P-EMB and Poisson PCA parameterize it multiplicatively. For the EF-EMB models and Poisson PCA, we use stochastic optimization with $\ell_2$ regularization. For HPF, we use variational inference. See Table 5 in Supplement C for details.

**Evaluation.** For the market basket data we hold out 5% of the trips to form the test set, also removing trips with fewer than two purchased different items. In the MovieLens data we hold out 20% of the ratings and set aside an additional 5% of the non-zero entries from the test for validation. We report prediction performance based on the normalized log-likelihood on the test set. For P-EMB and AP-EMB, we compute the likelihood as the Poisson mean of each nonnegative count (be it a purchase quantity or a movie rating) divided by the sum of the Poisson means for all items, given the context. To evaluate HPF and Poisson PCA at a given test observation we recover the factor loadings using the other test entries we condition on, and we use the factor loading to form the prediction.

**Predictive performance.** Table 3 summarizes the test log-likelihood of the four models, together with the standard errors across entries in the test set. In both applications the P-EMB model outperforms HPF and Poisson PCA. On shopping data P-EMB with $K = 100$ provides the best predictions; on MovieLens P-EMB with $K = 20$ is best. For P-EMB on shopping data, downweighting the contribution of the zeros gives more accurate estimates.

**Item similarity in the shopping data.** Embedding models can capture qualitative aspects of the data as well. Table 4 shows four example products and their three most similar items, where similarity is calculated as the cosine distance between embedding vectors. (These vectors are from P-EMB with downweighted zeros and $K = 100$.) For example, the most similar items to a soda are other sodas; the most similar items to a yogurt are (mostly) other yogurts.

The P-EMB model can also identify complementary and substitutable products. To see this, we compute the inner products of the embedding and the context vectors for all item pairs. A high value of the inner product indicates that the probability of purchasing one item is increased if the second item is in the shopping basket (i.e., they are complements). A low value indicates the opposite effect and the items might be substitutes for each other.

We find that items that tend to be purchased together have high value of the inner product (e.g., potato chips and beer, potato chips and frozen pizza, or two different types of soda), while items that are substitutes have negative value (e.g., two different brands of pasta sauce, similar snacks, or soups from different brands). Other items with negative value of the inner product are not substitutes, but they are rarely purchased together (e.g., toast crunch and laundry detergent, milk and a toothbrush). Supplement D gives examples of substitutes and complements.

**Topics in the movie embeddings.** The embeddings from MovieLens data identify thematically similar movies. For each latent dimension $k$, we sort the context vectors by the magnitude of the $k$th component. This yields a ranking of movies for each component. In Supplement E we show two example rankings. (These are from a P-EMB model with $K = 50$.) The first one contains children's movies; the second contains science-fiction/action movies.

### Acknowledgments

This work is supported by the EU H2020 programme (Marie Skłodowska-Curie grant agreement 706760), NFS IIS-1247664, ONR N00014-11-1-0651, DARPA FA8750-14-2-0009, DARPA N66001-15-C-4032, Adobe, the John Templeton Foundation, and the Sloan Foundation.

## Footnotes

[1] One might be tempted to see this as a probabilistic model that is conditionally specified. However, in general it does not have a consistent joint distribution (Arnold et al., 2001).

[2]We also analyzed unlagged data but all methods resulted in better reconstruction on the lagged data.

[3]We thank IRI for making the data available. All estimates and analysis in this paper, based on data provided by IRI, are by the authors and not by IRI.

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
