[Supplementary Material · supplement.pdf]

# Supplement to Exponential Family Embeddings

## A  Inference

We fit the embeddings $\rho[i]$ and context vectors $\alpha[i]$ by maximizing the objective function in Equation (3). We use stochastic gradient descent (SGD).

We first calculate the gradient, using the identity for exponential family distributions that the derivative of the log-normalizer is equal to the expectation of the sufficient statistics, i.e., $\mathbb{E}[t(X)] = \nabla_\eta a(\eta)$. With this result, the gradient with respect to the embedding $\rho[j]$ is

$$\nabla_{\rho[j]}\mathcal{L} = \sum_{i=1}^{I} \big(t(x_i) - \mathbb{E}[t(x_i)]\big)\nabla_{\rho[j]}\eta_i + \nabla_{\rho[j]}\log p(\rho[j]). \tag{4}$$

The gradient with respect to $\alpha[j]$ has the same form. In Supplement B, we detail this expression for the particular models that we study empirically (Section 3).

The gradient in Equation (4) can involve a sum of many terms and be computationally expensive to compute. To alleviate this, we follow noisy gradients using SGD. We form a subsample $\mathcal{S}$ of the $I$ terms in the summation, i.e.,

$$\widehat{\nabla}_{\rho[j]}\mathcal{L} = \frac{I}{|\mathcal{S}|}\sum_{i\in\mathcal{S}} \big(t(x_i) - \mathbb{E}[t(x_i)]\big)\nabla_{\rho[j]}\eta_i + \nabla_{\rho[j]}\log p(\rho[j]), \tag{5}$$

where $|\mathcal{S}|$ denotes the size of the subsample and where we scaled the summation to ensure an unbiased estimator of the gradient. Equation (5) reduces computational complexity when $|\mathcal{S}|$ is much smaller than the total number of terms. At each iteration of SGD we compute noisy gradients with respect to $\rho[j]$ and $\alpha[j]$ (for each $j$) and take gradient steps according to a step-size schedule. We use Adagrad (Duchi et al., 2011) to set the step-size.

**Relation to negative sampling.**  In language, particularly when seen as a collection of binary variables, the data are sparse: each word is one of a large vocabulary. When modeling sparse data, we split the sum in Equation (4) into two contributions: those where $x_i > 0$ and those where $x_i = 0$. The gradient is

$$\nabla_{\rho[j]}\mathcal{L} = \sum_{i:x_i>0} \big(t(x_i) - \mathbb{E}[t(x_i)]\big)\nabla_{\rho[j]}\eta_i + \sum_{i:x_i=0} \big(t(0) - \mathbb{E}[t(x_i)]\big)\nabla_{\rho[j]}\eta_i \tag{6}$$
$$+ \nabla_{\rho[j]}\log p(\rho[j]).$$

We compute the first term of the gradient exactly—when the data is sparse there are not many summations to make—and we estimate the second term with subsampling. Compared to computing the full gradient, this reduces the complexity when most of the entries $x_i$ are zero. But, it retains the strong information about the gradient that comes from the non-zero entries.

This relates to negative sampling, which is used to approximate the skip-gram objective (Mikolov et al., 2013b). Negative sampling re-defines the skip-gram objective to distinguish target (observed) words from randomly drawn words, using logistic regression. The gradient of the stochastic objective is identical to a noisy but biased estimate of the gradient in Equation (6) for a Bernoulli embedding (B-EMB) model. To obtain the equivalence, preserve the terms for the non-zero data and subsample terms for the zero data. While an unbiased stochastic gradient would rescale the subsampled terms, negative sampling does not. It is thus a biased estimate, which down-weights the contribution of the zeros.

## B  Stochastic Gradient Descent

To specify the gradients in Equation 4 for the SGD procedure we need the sufficient statistic $t(x)$, the expected sufficient statistic $\mathbb{E}[t(x)]$, the gradient of the natural parameter with respect to the embedding vectors and the gradient of the regularizer on the embedding vectors. In this appendix we specify these quantities for the models we study empirically in Section 3.

## B.1 Gradients for Gaussian embedding (G-EMB)

Using the notation $i = (n, t)$ and reflecting the embedding structure $\rho[i] = \rho_n$, $\alpha[i] = \alpha_n$, the gradients with respect to each embedding and each context vector becomes

$$\nabla_{\rho_n} \mathcal{L} = -\lambda \rho_n + \frac{1}{\sigma^2} \sum_{t=1}^{T} \left( x_{(n,t)} - \rho_n^\top \sum_{m \in c_n} x_{(m,t)} \alpha_m \right) \left( \sum_{m \in c_n} x_{(m,t)} \alpha_m \right) \tag{7}$$

$$\nabla_{\alpha_n} \mathcal{L} = -\lambda \alpha_n + \frac{1}{\sigma^2} \sum_{t=1}^{T} \sum_{m|n \in c_m} \left( x_{(m,t)} - \rho_m^\top \sum_{r \in c_m} x_{(r,t)} \alpha_r \right) \left( x_{(n,t)} \rho_m \right) \tag{8}$$

## B.2 Gradients for nonnegative Gaussian embedding (NG-EMB)

By restricting the parameters to be nonnegative we can learn nonnegative synaptic weights between neurons. For notational simplicity we write the parameters as $\exp(\rho)$ and $\exp(\alpha)$ and update them in log-space. The operator $\circ$ stands for element wise multiplication. With this notation, the gradient for the NG-EMB can be easily obtained from Equations 7 and 8 by applying the chain rule.

$$\nabla_{\rho_n} \mathcal{L} = -\lambda \exp(\rho_n) \circ \exp(\rho_n) \tag{9}$$
$$+ \frac{1}{\sigma^2} \sum_{t=1}^{T} \left( x_{(n,t)} - \exp(\rho_n)^\top \sum_{m \in c_n} x_{(m,t)} \exp(\alpha_m) \right) \left( \sum_{m \in c_n} x_{im} \exp(\rho_n) \circ \exp(\alpha_m) \right)$$

$$\nabla_{\alpha_n} \mathcal{L} = -\lambda \exp(\alpha_n) \circ \exp(\alpha_n) \tag{10}$$
$$+ \frac{1}{\sigma^2} \sum_{t=1}^{T} \sum_{m|n \in c_m} \left( x_{(m,t)} - \exp(\rho_m)^\top \sum_{r \in c_m} x_{(r,t)} \exp(\alpha_r) \right) \left( x_{(n,t)} \exp(\rho_m) \circ \exp(\alpha_n) \right)$$

## B.3 Gradients for Poisson embedding (P-EMB)

We proceed similarly as for the G-EMB model.

$$\nabla_{\rho_n} \mathcal{L} = -\lambda \rho_n + \sum_{t=1}^{T} \left( x_{(n,t)} - \exp\left( \rho_n^\top \sum_{m \in c_n} x_{(m,t)} \alpha_m \right) \right) \left( \sum_{m \in c_n} x_{(m,t)} \alpha_m \right) \tag{11}$$

$$\nabla_{\alpha_n} \mathcal{L} = -\lambda \alpha_n + \sum_{t=1}^{T} \sum_{m|n \in c_m} \left( x_{(m,t)} - \exp\left( \rho_m^\top \sum_{r \in c_m} x_{(r,t)} \alpha_r \right) \right) \left( x_{(n,t)} \rho_m \right) \tag{12}$$

## B.4 Gradients for additive Poisson embedding (AP-EMB)

Here, we proceed in a similar manner as for the NG-EMB model.

$$\nabla_{\rho_n} \mathcal{L} = -\lambda \exp(\rho_n) \circ \exp(\rho_n) + \sum_{t=1}^{T} \left( \frac{x_{(n,t)}}{\rho_n^\top \sum_{m \in c_n} x_{(m,t)} \alpha_m} - 1 \right) \left( \sum_{m \in c_n} x_{(m,t)} \alpha_m \right) \tag{13}$$

$$\nabla_{\alpha_n} \mathcal{L} = -\lambda \exp(\alpha_n) \circ \exp(\alpha_n) + \sum_{t=1}^{T} \sum_{m|n \in c_m} \left( \frac{x_{(m,t)}}{\rho_m^\top \sum_{r \in c_m} x_{(r,t)} \alpha_r} - 1 \right) \left( x_{(n,t)} \rho_m \right) \tag{14}$$

## C  Algorithm Details

| | Model | minibatch size | regularization parameter | number iterations | negative samples |
|---|---|---|---|---|---|
| neuro | G-EMB | 100 | 10 | 500 | n/a |
| neuro | NG-EMB | 100 | 0.1 | 500 | n/a |
| shopping | all models | n/a | 1 | 3000 | 10 |
| movies | all models | n/a | 1 | 3000 | 10 |

**Table 5:** Algorithm details for the models studied in Section 3.

## D  Complements and Substitutes in the Shopping Data

Table 6 shows some pairs of items with high inner product of embedding vectors and context vector. The items in the first column have higher probability of being purchased if the item in the second column is in the shopping basket. We can observe that they correspond to items that are frequently purchased together (potato chips and beer, potato chips and frozen pizza, two different sodas).

Similarly, Table 7 shows some pairs of items with low inner product. The items in the first column have lower probability of being purchased if the item in the second column is in the shopping basket. We can observe that they correspond to items that are rarely purchased together (detergent and toast crunch, milk and toothbrush), or that are substitutes of each other (two different brands of snacks, soup, or pasta sauce).

| Inner product | Item 1 | Item 2 |
|---|---|---|
| 2.12 | Diet 7 Up lemon lime soda | Diet Squirt citrus soda |
| 2.11 | Old Dutch original potato chips | Budweiser Select 55 Lager beer |
| 2.00 | Lays potato chips | DiGiorno frozen pizza |
| 2.00 | Coca Cola zero soda | Coca Cola soda |
| 1.99 | Soyfield vanilla organic yogurt | La Yogurt low fat mango |

**Table 6:** Market basket: List of several of the items with high inner product values. Items from the first column have higher probability of being purchased when the item in the second column is in the shopping basket.

| Inner product | Item 1 | Item 2 |
|---|---|---|
| −5.06 | General Mills cinnamon toast crunch | Tide Plus liquid laundry detergent |
| −5.00 | Doritos chilli pepper | Utz cheese balls |
| −5.00 | Land O Lakes 2% milk | Toothbrush soft adult (private brand) |
| −5.00 | Beef Swanson Broth soup 48oz | Campbell Soup cans 10.75oz |
| −4.99 | Ragu Robusto sautéed onion & garlic pasta sauce | Prego tomato Italian pasta sauce |

**Table 7:** Market basket: List of several of the items with low inner product values. Items from the first column have lower probability of being purchased when the item in the second column is in the shopping basket.

# E   Movie Rating Results

Tables 8 and 9 show clusters of ranked movies that are learned by our P-EMB model. These rankings were generated as follows. For each latent dimension $k \in \{1, \cdots, K\}$ we sorted the context vectors according their value in this dimension. This gives us a ranking of context vectors for every $k$. Tables 8 and 9 show the 10 top items of the ranking for two different values of $k$. Similar as in topic modeling, the latent dimensions have the interpretation of topics. We see that sorting the context vectors this way reveals thematic structure in the collection of movies. While Table 8 gives a table of movies for children, Table 9 shows a cluster of science-fiction and action movies (with a few outliers).

| # | Movie Name | Year | Rank |
|---|------------|------|------|
| 1 | Winnie the Pooh and the Blustery Day | 1968 | 0.62 |
| 2 | Cinderella | 1950 | 0.50 |
| 3 | Toy Story | 1995 | 0.46 |
| 4 | Fantasia | 1940 | 0.44 |
| 5 | Dumbo | 1941 | 0.43 |
| 6 | The Nightmare Before Christmas | 1993 | 0.37 |
| 7 | Snow White and the Seven Dwarfs | 1937 | 0.37 |
| 8 | Alice in Wonderland | 1951 | 0.35 |
| 9 | James and the Giant Peach | 1996 | 0.35 |

**Table 8:** Movielens: Cluster for "kids movies".

| # | Movie Name | Year | Rank |
|---|------------|------|------|
| 1 | Die Hard: With a Vengeance | 1995 | 1.25 |
| 2 | Stargate | 1994 | 1.19 |
| 3 | Star Trek IV: The Voyage Home | 1986 | 1.14 |
| 4 | Manon of the Spring (Manon des sources) | 1986 | 1.14 |
| 5 | Fifth Element, The | 1997 | 1.14 |
| 6 | Star Trek VI: The Undiscovered Country | 1991 | 1.13 |
| 7 | Under Siege | 1992 | 1.11 |
| 8 | GoldenEye | 1995 | 1.07 |
| 9 | Supercop | 1992 | 1.07 |

**Table 9:** Movielens: Cluster for "science-fiction/action movies".