[Reviews · NeurIPS 2016]

Reviewer 1

Summary

This paper presents exponential family embeddings, which are an extension of word embeddings to more general types of data (continuous, count). The two main ingredients are (1) a context for each observation, which is the set of related observations to be conditioned on, and (2) an embedding, which governs the conditional distribution of the data. These conditional distributions are modeled as exponential families, where the natural parameter goes through a link function as in generalized linear models. Results are shown on neuron calcium data, movie preference data, and market basket data.

Qualitative Assessment

* This is a simple idea, but a good one. The paper is clearly written. * The main problem I have is that the results are mostly qualitative, especially the last page (8) of the writing. The analysis stops at inferring the embedding features, and it would be made stronger if it was shown how these could be used in a further analysis or task. That being said, I think there is just enough quantitative results that are convincing enough. * I think some details are missing on the regularization being used in the various examples. There is mention of l_2 regularization in the first two examples. But the weight (Gaussian precision) on the regularization is not specified (or I could not find it). How is this tuned and how sensitive are the results to the regularization? Example 4 does not mention regularization (does it have any?). * It is not clear how the result on the neural data would be useful in practice. Particularly, how does Figure 1 help someone understand the data better?

Confidence in this Review

2-Confident (read it all; understood it all reasonably well)


Reviewer 2

Summary

The authors present a new class of models (exponential family embeddings, EF-EMB) that generalize word embeddings to other types of high dimensional data. The core idea is that data are influenced by other data within the same context (other data points selected by some measure of relatedness). For the purpose of embeddings, the authors borrow ideas from exponential families and generalized linear models. Efficient algorithms are developed and (existing) special cases are discussed. Results are presented on three different types of data.applications, namely, neuroscience (continuous), shopping (counts) and movie ratings data (ordinal). EF-EMB is a recipe with three ingredients: context, conditional exponential family and embedding structure. Contexts consist of subsets of data, are modelers' choice and depend on the application. Conditional exponential families are essentially generalized linear models with context as covariates. The embedding structure, which is key, determines how contexts are shared. The authors present examples of Gaussian, nonnegative Gaussian, Poisson, additive Poisson, Multinomial and Bernoulli embeddings. Parameters (embeddings and contexts) are estimated via stochastic gradient descent.

Qualitative Assessment

In general terms I find the approach sound and clearly described, and commend the authors for making clear that the proposed model is not a probabilistic model in the strict sense. Although the approach is not too innovative I can see it may be useful in some scenarios where context is self-specified by the structure of the data. The results in Table 1 are convincing, however, I would like to see the authors commenting on cases where the proportion of held out neurons is larger than 25% or in the presence of missing data, in which case one may be inclined to think that FA may perform as good or even better than EF-EMB. About the results for the second experiment shown in Table 2, I would like to see error bars as in Table 1 and some intuition on why HPF performs so poorly on market data and why AP-EMB performs so poorly on the moving ratings data.

Confidence in this Review

2-Confident (read it all; understood it all reasonably well)


Reviewer 3

Summary

This paper extends the idea of word embeddings from text to other high dimensional data. This is done by replacing surrounding words by a dataset-dependent context, which could be surrounding neurons in a neural recording, or other items in the shopping cart of shopper. The conditional distribution of the data given its context is then modeled as an exponential family, e.g. Poisson for categorical data or Gaussian for continuos. This leads to an objective function that describes a GLM which can be optimized using gradient descent.

Qualitative Assessment

Noise Contrastive Estimation should cite the original work, Gutmann and Hyvarinen 2010 (JMLR). In section 2.2, please elaborate on the relationship of the "summation over many terms" with the partition function that is approximated using NCE in previous word embedding models. It's not clear from the manuscript or supplement A what the sum over capital I refers to. I would like to see the relationship to previous word embedding models explored more fully, and how the subsampling objective relates to other approximations used in previous work.

Confidence in this Review

2-Confident (read it all; understood it all reasonably well)


Reviewer 4

Summary

This paper presents a general approach for learning exponential family embeddings. The motivation is that in various applications, we are interested in capturing the relationship of a data point to other data points that occur in its context. For example, in the case of word embeddings, the goal is to learn the conditional probability of a word given its context words. The authors use different distributions from the exponential family to learn embeddings for different datasets, for example, read-valued neural data and discrete market basket data. Because all these models use distributions from the exponential family, they share the same general objective function and gradient formulations. (For each specific application, the context and conditional probabilities are defined.) The proposed models outperform the existing approaches based on factor analysis on both the neural data and market basket datasets.

Qualitative Assessment

The paper is well-written and clear. The most interesting aspect of the work is that it provides a general approach for learning embeddings for any dataset that can be modeled with an exponential family distribution: Although similar approaches for learning embeddings for words have been proposed, this paper provides a general framework that can be applied to various type of data (not just language). However, the model is only suitable for cases that all the data points in the context (of a given data point) are considered equally relevant, and thus cannot capture the finer-grained relations between data point pairs. An interesting direction for the work would be extending the evaluation of the shopping and movie rating data: to what extent the predictions of the model (for example, items that are more/less likely to be purchased together) improve an external application?

Confidence in this Review

2-Confident (read it all; understood it all reasonably well)


Reviewer 5

Summary

This paper proposes a novel embedding method called exponential family embedding (EF-EMB). EF-EMB can deal with several types of data over the previous embedding method. Some stochastic gradient method is proposed to estimate the parameters in EF-EMB.

Qualitative Assessment

This paper is well-written, and includes a couple of examples where the proposed method could be useful. My simple question is about experiments. The proposed method is compared with only conventional methods such as factor analysis. I am not sure that this is appropriate comparison to demonstrate the superiority of the proposed method.

Confidence in this Review

1-Less confident (might not have understood significant parts)


Reviewer 6

Summary

The paper deals with the general problem of finding latent embeddings for a variety of problems and places the problem in a Bayesian framework by showing that embedding problems from several different applications can be expressed as inference in an exponential family scenario. In this case each observation has both an embedding and a context vector. This framework is defined by three main choices (1) defining the context of an observation, (2) deciding how the conditional probability of an observed data point is linked to its context and (3) how the latent embeddings and context vectors are shared across different observations. Inferring the embeddings and context vector can then be done via stochastic gradient descent fairly simply due to properties of the gradient when dealing with exponential families The generality of this framework is then demonstrating that four different real-life embeddings problems can be formulated in this way. The three choices are addressed for each of the four and a gradient formula is derived for each. In the case of two of the scenarios (real valued neural data and Poisson observations for shopping/movies) the EF-EMB method is applied to some real life data sets and empirically strong results are shown in terms of likelihood estimations/errors. Moreover, some quantitative results on these datasets are given to show how having both a context vector and latent embedding can help in the interpretation of results in these contexts.

Qualitative Assessment

Technical quality: 3 The experimental method was well motivated and correctly seemed to generalize to the scenarios given. The derived gradients were correct (if a little too succiently explained for my taste). The experimental results themselves were strong (though I am not an expert in any of the areas so hard to say whether they picked the exact best things to compare against) and the additional quantitate analysis was a good decision since it shows there will be a consistent way to interpret results. I think the method seems to actually produce good results, has a good intuition behind it and could actually be useful which in my mind is much more important than excessively long proofs. There were a few questions about the experiment analysis that should have been addressed. In table 2 it’s odd for example that both P-EMB and AP-EMB both perform worse from K=50 to K=100. Since this measures purely the likelihood and does not include the prior theoretically increasing the size of K should only ever improve the score. In the worst case, letting the last 50 embedding and context vector values always be zero and letting the first 50 be the same as the optimal result for K=50 should give you a score as before. This seems to suggest that the SGD has some problems converging for higher dimensions so it would have been nice to have this addressed (e.g: was the convergence stopped due to time constraints). Generally mentioning the running times of EF-EMB methods versus the alternatives would have also been informative. One aspect I found confusing is that the link function introduced in equation 2 was generally assumed to be the identity function. However this runs into problems when the size of the context for each observation point is not constant (as was noted in the experimental analysis of the movie ratings). It seems like taking the dot product of the latent embedding and the weighted average of the context vectors is more generally the correct option. Novelty/originality: 3 There is no new algorithm here but the idea of putting many different embeddings under the same sort of framework is new. The main novel contribution is noting that this could be done and esthablishing which key steps are required to do this. Moreover, each of the 4 exponential family versions of the embeddings for the difference scenarios are a novel contribution and appear to be empirically useful formulations. The experimental methods are fairly standard but seem fully appropriate for the problem of comparing old methods with EF-EMB. Potential Impact: 4 I feel the authors have esthablished several key things about this framework. Firstly they have shown that it is fairly general through good use of 4 very different examples which seem well served by the method. Secondly, they have shown that the method is fairly easy to apply to a problem if you are familiar with the problem and exponential families. Once I review EF it was fairly easy to follow how they derived these new models and this ease of understanding makes it likely to use. Finally, they have shown that this is useful both in terms of the quantitate results but also in terms of giving us some examples of how this can be used to interpret results for exploratory reasons. Though this third part is perhaps the one I would like to see improved the most since they have not talked about how this fares for problem where the dimensionality of the embedding is very high (K was limited to just 100 in the examples) nor shown results for the more well studied problem of word embeddings. As previously mentioned there is no mention of how long it takes to run compared to other options which is also something that would help us understand how applicable the method might be. Clarity and presentation: 3 Disclaimer, I last looked at exponential families 5 or so years ago so had to refresh myself on this several times so keep that in mind when reading this section. In my mind this was the weakest area of the paper and also the least balanced. I though the paper was very well motivated as we moved through sections 1 and 2: I was following the idea well, understood the idea of what needs to be defined to make an EF-EMB etc. Things starting getting a little harder to follow around 2.1. I think the authors presume a fair amount of familiarity with exponential families and would have benefitted from just stating things a little more explicitely. In example 1 of 2.1 for example there is discussion of a Gaussian with mean esthablished by the inner product but no mention of the standard deviation: it was only after a bit of reading that I saw this is replaced by lambda in the case of an exponential family prior-posterior case. The hardest aprt to follow was 2.2. Without referencing the canonical way to set up an exponential family conditional ( f(x| theta ) = h(x)exp(eta(theta). T(x) – A(theta) ) its not clear what terms like eta represent or to even find out what a(eta) is (which appears without warning in equation three first). Obviously its not a huge flaw but it does add a significant amount of effort to what was otherwise a pretty smooth read beforehand. I was also a little confused by the use of “regularizer” in the objective function. From what I can understand this could more properly defined as the priors of the latent embeddings and context vectors and ensures that the objective function is seeking to maximize the entire posterior. Some of the equations’ derivations were a little terse. I can understand that in the interest of space not mentioning them in the main paper but it would have been nice to have a little discussion of what the prior is and how the chain rule is used in appendix A.1 for example. Again, nothing that can’t be worked out without a little extra research/calculations but neither is it something desirable. The experimental analysis was fairly easy to follow but I would have liked a little more detail on exactly what kind of factor analysis was being used for the first set of experiments as well as an explaination of why other parameter choices (such as higher values of k) were not explored. Summary Overall though I rather enjoyed reading this paper and think it is a great piece of work. It provides a framework for several ideas that is theoretically justified, useful and somewhat interpretable. The fact that it is fairly to simple to understand and does not rely on advanced mathematical machinery is yet another advantage. The generalizations seem generally good though there is certainly space for more investigation (e.g: can it be modified to admit cases where context data points can be weighted, e.g weighing the context word right next to the target word more than that five places to the right). The experimental work certainbly looks promising but I would like to see that get filled in with more analysis: especially on more computationally intensive scenarios.

Confidence in this Review

2-Confident (read it all; understood it all reasonably well)